# Examining the "Natural Resource Curse" and the Impact of Various Forms of Capital in Small Tourism and Natural Resource-Dependent Economies

**Petar Kurecic** [1,*] **and Filip Kokotovic** [2]

1    Department of Business and Economics, University North, 42000 Varazdin, Croatia
2    University College of International Relations and Diplomacy Dag Hammarskjöld, 10000 Zagreb, Croatia; filip.kokotovic@hotmail.com
*    Correspondence: pkurecic@unin.hr; Tel.: +385-98-980-8577

**Abstract:** The problem of the relevance of human and natural capital, as well as the potential adverse effect of natural capital on economic growth, has gained increased attention in development economics. The aim of this paper is to assess, theoretically and empirically, the relevance of several forms of capital on economic growth in certain small economies that are dependent upon tourism or natural resources. The empirical framework is based on Impulse Response Functions obtained from Vector Autoregressive models in which we focus on the model where economic growth is the dependent variable for ten small economies that are dependent upon either tourism or natural resources. We find that there is evidence of the "natural resource curse", especially in the economies that have a strong dependence on resources that are easily substitutable and whose prices constantly fluctuate. We further find that in the majority of observed cases, the type of capital these small economies are most dependent on for their economic growth causes negative impulses in the majority of the observed periods. Therefore, the main policy recommendation should be to assure that even these small economies should strive towards further diversification and avoid dependence on only one segment of their economy.

**Keywords:** natural capital; human capital; economic growth; small economies; Vector Auto regression; natural resource curse

**JEL Classification:** O47; Q32; E32; I25; O44

## 1. Introduction

There is a well-known differentiation between different forms of capital: natural, financial, foreign, real, human, and social (Gylfason, 2006 [1]). Goodwin (2003 [2], p. 1) on the other hand differentiates between five kinds of capital: financial, natural, produced, human, and social, stating: "All are stocks that have the capacity to produce flows of economically desirable outputs. The maintenance of all five kinds of capital is essential for the sustainability of economic development." All these forms of capital represent determinants of economic growth. The difference is how these different kinds of capital affect economic growth and is this relationship statistically significant or not. Regarding natural capital, there are several definitions present in the literature. Nevertheless, measuring of the natural capital stock poses a problem in research, as for example England (1998 [3]), as well as Pearce and Turner (1990 [4]) elaborate. Natural capital, according to Daly (1994 [5]), is the stock that yields a flow of natural services and tangible natural resources. He mentions fossil fuel reserves and populations of fish and trees. For Berkes and Folke (1994 [6]), natural capital consists of three major components:

Non-renewable resources extracted from ecosystems, renewable resources produced and maintained by ecosystems, and environmental services (in [3], p. 258). According to van der Ploeg (2011 [7], p. 372) natural capital consists of subsoil assets, timber resources, nontimber forest resources, protected areas, cropland, and pastureland. Due to data problems, fisheries, subsoil water, and diamonds are excluded. The explicit value of ecosystems is not evaluated either. The value of natural capital is estimated from world prices and local costs.

Measuring natural capital poses several problems, and one of those is the endogeneity of measures. Therefore, it is very difficult, if not impossible, to use a truly exogenous variable, as Torvik (2009 [8], pp. 244–245) elaborates: "What is a measure of natural resources that is truly exogenous? Finding such a measure would allow a natural experiment on the causal effect from resource abundance to growth. To my knowledge, there exists no paper that has come up with a true exogenous measure of resource abundance."

Torvik ([8], p. 246) also addresses the problem of calculating the change of natural capital into other forms of capital: "When a country sells its oil and puts the proceeds in the financial market, it reduces the natural capital of the country while it increases the financial capital of the country. The wealth of the country is unchanged. Should it happen to consume all the proceeds from the sale of oil, the correct understanding is that its savings rate is negative."

The share of produced assets in total wealth is more or less the same irrespective of how poor or rich a country is. However, the share of natural capital in total wealth is much higher in poorer countries while the share of intangible capital in total wealth is substantially higher in richer economies. Interestingly, richer countries have a substantially higher value of natural capital per capita despite having lower shares of natural capital in total wealth ([7], p. 372). This can be connected with the processes of depleting natural capital rapidly in the poor countries. Hence in the discourse on sustainable development, constant "natural capital" is frequently referred to as a criterion for sustainability, and the constant "natural capital" condition is normally called "strong sustainability" (Hinterberger [9], p. 3) it can be concluded that poor countries that are heavily dependent on natural capital have a very weak sustainability, and the only chance for increasing sustainability is the development of forms of capital other than natural.

The main intention of this paper is to study the research hypotheses empirically, by comparing the role of both natural and human capital in the small economies that are dependent on natural resource exports or tourism (economies that primarily depend on foreign guests and consumption, i.e. receptive tourist countries), respectively. We absolutely recognize the differences between natural resource extraction and tourism, as two very different types of economic activities. However, some similarities occur, which have led us to believe that resource export dependent countries and countries dependent on tourism should be compared, regarding the causes of vulnerability of their economies and long-term growth perspectives. Natural resource extraction (in this context we primarily refer to fuels and/or minerals and ores) and export, if not accompanied by industrial production in the countries of origin, in most cases cause economic stagnation or even recession, rising social inequality, keep or increase widespread poverty, rise corruption, hinder democratic processes and "kill" political freedom and civil liberties. That is what various researchers of the "resource curse", natural capital and the role of genuine savings (among other related phenomena) on country (through cross-country comparative analyses) ([7,8]; Atkinson and Hamilton, 2003 [10]; Hamilton and Atkinson, 2006 [11]), and regional (through subnational analyses) level (Fleming et al, 2015 [12]; as well as Fleming and Measham, 2015 [13]) elaborate. There are only a couple of exceptions to that rule, hence actually verifying it. The best exception due to the cultural and historical patterns of social and economic development is most probably Norway. Exceptions due to the planned government policies of economic transformation include Botswana, as well as Qatar and Dubai, which in the recent two decades have been able to diversify their economies. A detailed explanation of the resource curse phenomenon is given in the discussion section.

The effect of human capital on economic growth has already been considered in many studies. Indeed, since this issue first caught the attention of researchers, several significant improvements have been made. An issue that has caused a significant debate is the use of human capital proxies in order to assess their relevance for economic growth. Notably the work of Barro (1991 [14]) has made it perfectly clear that his variables are human capital proxies. These original considerations have meanwhile been reviewed, and Judson (2002 [15]) notes that the commonly used proxy variables are education expenditure, years spent in education or education enrolment. The relevance of the variable that Judson attempted to construct displays the sharp need to redefine the measurement of human capital.

This study approaches two highly relevant issues from a similar approach. Most notably, this study does not question the assertion that human capital proxy variables are imperfect and provide at times disputable results. All variables are to a different degree imperfect. An issue that should be pointed out is that the vast majority of variables that are used in econometric research are to at least some degree arbitrary. Many researchers would agree that GDP per capita is far from a perfect indicator of the quality of life. However, several have noted that GDP itself is far from a perfect measurement of economic growth, and yet today it is practically used as a synonym for it (Costanza, et al., 2009 [16]). The use of various research methods and approaches should be considered beneficial to the overall growth of science rather than assuming a dogmatic approach.

The main goal of this study is to critically analyse the similarities and differences between the relevance of human capital and natural capital in small economies that are affected by the "natural resource curse" in comparison to countries that are dependent upon tourism as a source of revenue. The main idea behind this approach is that these countries suffer from many similar issues. Due to their size and properties, most of them are also highly dependent on one or two sources of revenue, which through a multiplicative effect then has a crucial effect on their economies. This can be highly detrimental to their economies, especially taking into account the fact that factors such as oil prices and the arrival of tourists are factors that these small economies cannot hope to control or even prepare for.

We are aware of the weak points of cross-country comparative studies ([12,13]; Borge, et al., 2015 [17]). Fleming, et al. ([12], p. 626) point out: "Even in cases when countries obtain national gains from resources, subnational negative economic growth effects—a regional resource curse—can emerge in regions surrounding extractive industries."

The aforementioned authors also state that the resource curse occurs from some of the hypotheses of the curse, which they have listed operating regionally, but it can also occur from the labour demand shock that mining expansion produces in resource-rich regions and its neighbours.

It is also important to note the other reason for performing regional level studies: "Within-country analyses of the resource curse provide more robust evidence than cross-country models as they reduce the unobserved heterogeneity generally given by institutional, cultural and historical background across countries" ([12], p. 629).

Due to these reasons, we have decided to perform an individual time series analysis of a number of small economies, to see if all analysed small economies show similar features regarding the dependence on natural resources or tourism. We believe that a scientific contribution can be made through the comparative analysis of a high level of dependence on natural resources exports or tourism (which is in the analysed economies dominantly foreign-originated, and therefore represents the export of services) experienced by selected small economies, respectively. Hence, only very small economies were analysed, the robustness of the results, because they were obtained only at the national level, is lower than it would be if we were to analyse large economies that show a certain level of resource exports dependence (Canada, Australia, Russia, etc.). Additionally, the unavailability of data at levels lower than national was also an insurmountable problem that we could not overcome if we were to analyse subnational levels, as the aforementioned authors have done in the cases of Australia or Norway, respectively.

## 2. Methodology

The smallness of the economy was taken as a prerequisite for inclusion into the research mainly because we wanted to test if the smallness of the economy itself is connected with a high dependence on natural resources rents which overlaps (in most cases) with significant economic underdevelopment and inability to develop other sectors of the economy. Nevertheless, smallness of the economy in most of the economies that hold these characteristics is not the cause, but the consequence of these characteristics, due to the low or extremely low level of GDP per capita. Therefore, smallness of the economy is not studied per se. It is primarily used as a filter that determines (in accordance with the aforementioned indicators) which economies are to be studied. The characteristics of studied resource-dependent economies are in accordance with the resource curse thesis (Humphreys, et al., 2007 [18]).

The current methodological constraints provide several limitations on the measurement of human and natural capital. As such, it is difficult to provide results that will provide absolute certainty, but a dogmatic approach is something researchers should strive to avoid. Many of the measurements are imperfect and this paper likewise suffers from similar limitations. Generally, when describing economic growth it is difficult not to mention the Cobb-Douglas function (Cobb and Douglas, 1928 [19]), as follows:

$$Y = AL^{\beta}K^{\alpha}, \tag{1}$$

where $\alpha$ and $\beta$ are output elasticities, $Y$ is the measurement of economic growth, $A$ is the total factor productivity, while $L$ and $K$ are respectively the labor and capital inputs. This function was criticized primarily for not taking into account different forms of capital, which is the key reason why today the majority of researchers favour the so-called Solow-Swan model and others. This model is partially based on the work of Solow (1939 [20]), while there are constant discussions whether Solow or Swan should be credited for the model we have today. This includes papers such as Diamond and Spencer (2008 [21]) that discuss the relevance of both the work of Solow and Swan to the model. Solow [20] begins with a function that does not differ significantly from the Cobb-Douglas function, and goes as follows:

$$Q = A(t)f(K,L), \tag{2}$$

where he takes into account the factors that affect the output ($Q$), based on the change of physical units of input of labor ($L$) and physical capital ($K$), as well as $A$, which accounts for the cumulated effects of change over time. The function of this paper is not to determine which researcher contributed more to the development of the function. Therefore, this paper refers to it as the Solow-Swan model. Perhaps the most significant change over time in regards to human capital was added in the research of Mankiw, Romer and Weil (1992 [22], p. 416), as follows:

$$Y(t) = K(t)^{\alpha}H(t)^{\beta}(A(t)L(t))^{1-\alpha-\beta}, \tag{3}$$

where the notable change in regards to the Solow-Swan model is the inclusion of the relevance of human capital on economic growth. Rather than taking any of these functions as dogmatic, this paper analyses the relevance of several variables that may influence economic growth, measured by the change of real GDP. An empiric evaluation of which factors influence economic growth will provide us with a more exact specification of the similarities or differences between the observed economies. Thus, this paper assumes that economic growth is significantly influenced by a series of factors that include several forms of capital, which can be simply described as:

$$Y(t) = f(K,L,t), \tag{4}$$

assuming that in the time period t economic growth is influenced by the output of labor ($L$) and capital ($K$). At this point, the discussion shifts to which best proxy variables can be used to describe

the change of economic growth and the relevant variables. In the case of the economies that this paper studies, this paper distinguishes several forms of capital whose measurement will be empirically assessed. Therefore, the model takes the form:

$$Y(t) = f(HC,\ NC,\ PC,\ L,\ t), \tag{5}$$

where we account for the existence of human, natural and physical capital that have different effects of economic growth in the observed time period $t$. This paper primarily analyses the relevance of the variables described in Table 1 towards the dependent variable, which is GDP measured in constant \$2010.

**Table 1.** Model specification.

| Variable | Abbreviation used | Source | Function |
|---|---|---|---|
| Gross Domestic Product | GDP | World Bank (2016) | Dependent variable |
| Tourism as percentage of GDP | Tourism | World Bank (2016), World Tourism Council (2016) | Independent variable |
| Total Natural Resource Rent | NC | World Bank (2016) | |
| Human capital stock | HC | Barro-Lee database (2013) | |
| Gross Domestic Product per capita | GDPpc | World Bank (2016) | |

The database originally created by Barro and Lee (2013 [23]) is perhaps the most recognized database that has measurements for long-term human capital stock. As the data is for 5-year averages, we use linear extrapolation in order to gain annual values. Although this approach has shortcomings in assuming that between two 5-year points the data moves in a precise linear trend, which is not correct, we find this approach preferable to using different human capital proxy variables such as school enrolment or education expenditure. This paper analyses the period of 1995–2014, for which data was available, for small tourism and natural resources dependent economies. The remaining data was extracted from the World Bank database (2016 [24]). Natural capital is measured through Gross Value Added (GVA).

Before implementing econometric models we use the test originally developed by and Dickey and Fuller (1979 [25]), with the lag length specification based on the information criterion introduced by Akaike (1974 [26]). This test has the null hypothesis of non-stationarity, meaning that the means and variances between the data are not equal. Stationarity is necessary in order to avoid errors of possible spurious, statistically insignificant results. Rejection of the null hypothesis of non-stationarity at the 5% significance level is necessary prior to conducting further econometric models or tests. As can be seen in Appendix A where we provide the full results of the unit root and other specification-related tests, for most of the countries the variables are integrated in different orders and several of them, mostly human capital (HC), are in several cases I(2).

The possible empirical approaches are however limited. As the variables are not cointegrated, the approach developed by Peseran and Shin (1999 [27]) can be implemented only if the variables are stationary or stationary in their first difference. Therefore, it seems most logical to capture the relationship between the variables using a Vector Autoregressive (VAR) model approach. This paper analyses each country individually rather than using a panel setting. The key reason is that the small economies that are observed by the paper have highly specific properties that might be more difficult to observe in a panel setting. Thus, we focus on the two out of four possible equations that such an approach provides us with the following:

$$\begin{aligned} GDP_t = \ & \alpha_0 + \alpha_1 GDP_{t-1} + \ldots + \alpha_{1,2} GDP_{t-n} + \alpha_2 GFCF_{t-1} + \ldots + \alpha_{2,1} GFCF_{t-n} \\ & + \alpha_3 HC_{t-1} + \ldots + \alpha_{3,1} HC_{t-n} + \alpha_4 NC_{t-1} + \ \alpha_{4,1} NC_{t-n} + \ \varepsilon_{t,1}, \end{aligned} \tag{6.1}$$

$$
\begin{aligned}
HC_t \ = \ & \beta_0 + \beta_1 HC_{t-1} + \ldots + \beta_{1,2} HC_{t-n} + \beta_2 GDP_{t-1} + \ldots + GDP_{2,1} Y_{t-n} \\
& + \beta_3 GFCF_{t-1} + \ldots + \beta_{3,1} GFCF_{t-n} + \beta_4 NC_{t-1} + \ldots + \beta_4 NC_{t-n}\, \varepsilon_{t,2};
\end{aligned}
\tag{6.2}
$$

where the variables are abbreviated as specified in Table 1, $\alpha_0$ and $\beta_0$ are the constants, $\alpha_{1..4}$ and $\beta_{1..4}$ are the coefficients, $\varepsilon_{t,1,2}$ are the error terms in the maximum lag length $n$, determined based on the Akaike information criterion, in the time period $t$. The reason why we focus on these two specific equations is the research interest of this paper, which is determining the relationship between various forms of capital on economic growth, but we also include the question of whether natural capital has a significant effect on human capital. In order to examine the relationship we conduct Impulse Response Functions (IRF) that depend on the earlier work of Sims (1980 [28]).

Regarding the specification tests, the autocorrelation test is based on the research by Breusch (1978 [29]) as well as Godfrey (1978 [30]), the ARCH effect test is based on the work of Engle (1982 [31]), and the test for stability of the VAR inverse roots, which are essential for the stability of the model, based on Lütkepohl (1991 [32]).

The small economies for which data was available were initially selected as either resource-dependent or tourism-dependent, based on World Bank indicators [24] and data from the World Travel & Tourism Council [33]. There were certain data constraints, such as the limited period this paper could observe because the aforementioned source, as the most complete database, is available only for the period of 1995–2015. The research was focused on small economies of the world (classified by the size of their GDP) that are either highly dependent on natural resources rents as percentage of GDP or highly dependent on tourism as a sector of their economies. The negativities of small economies, besides other features, usually include vulnerability to external shocks (albeit resilience as well, especially in some cases), less opportunities to diversify the economy, smaller workforce base, small resource base (in most cases). A short discussion regarding such a specification, as well as some initial observations, is available in Appendix B. An important factor to note is that we apply a log transformation for GDP, HC and GDPpc. The average values for all of the countries observed, prior to the log transformation of the 3 previously mentioned variables, can be found in Table 2.

**Table 2.** Average values of variables 1995–2014.

| Country | GDP (in constant $2010) | Tourism (as percentage of GDP) | GVA (as percentage of GNI) | HC (stock of human capital) | GDPpc (in constant $2010) |
|---|---|---|---|---|---|
| The Gambia | 536,971,279.76 | 6.56 | −0.26 | 1,249.89 | 518.45 |
| Trinidad and Tobago | 12,012,729,470.09 | 3.34 | −6.29 | 2,463.22 | 9,847.27 |
| Lesotho | 1,206,862,818.90 | 4.86 | 18.26 | 1,719.04 | 701.54 |
| Belize | 711,605,918.56 | 11.03 | 9.98 | 2,150.78 | 3,027.54 |
| Guyana | 1,647,108,974.58 | 3.63 | 5.1 | 2,150.98 | 2,209.12 |
| Jamaica | 10,996,282,998.37 | 8.35 | 10.43 | 2,262.16 | 4,582.56 |
| Fiji | 2,293,187,461.92 | 12.59 | 1.57 | 2,430.64 | 3,075.16 |
| Liberia | 1,640,762,520.67 | 1.382 | −26.79 | 1,362.07 | 780.92 |
| Malawi | 3,781,976,645.782 | 2.825 | 7.96 | 1,410.4 | 384.81 |
| Barbados | 3,809,503,148.58 | 11.64 | 11.21 | 2,415.8 | 14,216.92 |

Source: Authors' calculations.

An important factor to emphasize is that such a categorization, into resource and tourism-dependent countries, does not influence the empirical segment of the paper hence each country is individually analysed. The calculations are conducted in the Gnu Regression, Econometrics and Time-series Library (GRETL).

## 3. Results

As specified in the methodological section, the first step is conducting unit root tests, all of the results present in the Appendix A.

### 3.1. VAR IRF Results

Based upon the results of the unit root test we may proceed to conduct further statistical analysis via the VAR models. The specification tests have determined that the models are not under the influence of the ARCH effect at the selected lag length; furthermore, they are not influenced by autocorrelation and have structurally stable parameters based upon the results of the results of the stability of the VAR inverse roots. All of these results are available in Appendix A, as well as the Cholesky ordering of the variables. After satisfying all these preconditions, it is possible to estimate and analyse the results that are provided by the IRFs.

The results for the Gambia, as presented in Figure 1, suggest that in none of the observed variables is there a significant long-term relationship. The model was estimated at the lag length 1, due to misspecification errors that occurred at two lags proposed by the Akaike information criterion. The initial response to tourism is slightly negative, but becomes statistically insignificant after two or three periods. The initial response of GDP to HC is strongly negative, yet it also becomes insignificant after four periods. The result of an impulse of NC on economic growth seems to cause a slightly negative effect, which does not persist after two periods. The initial reaction from HC to an impulse in NC seems to be completely neutral, which mostly persists throughout all 10 observed periods.

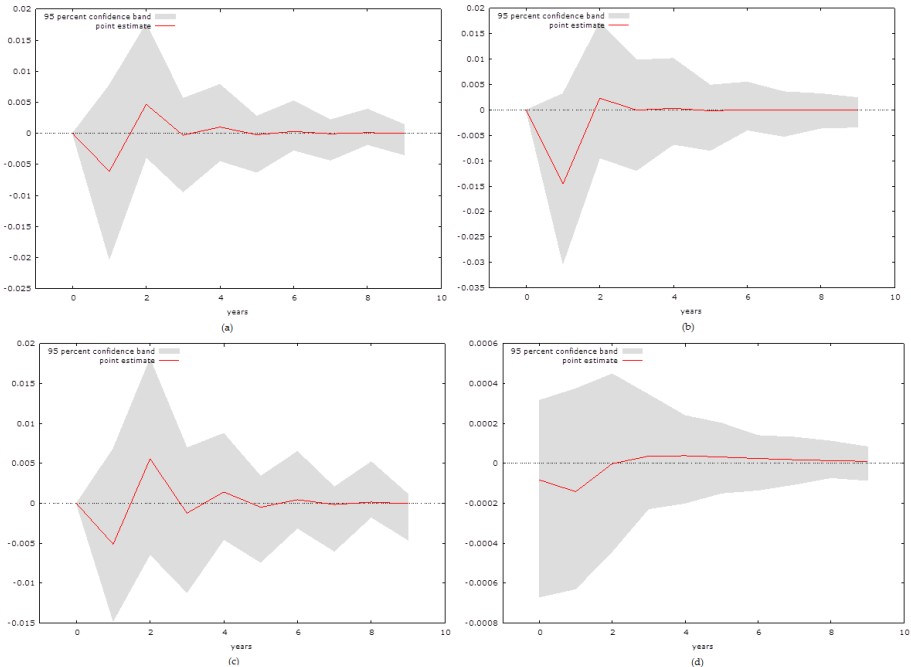

**Figure 1.** Impulse Response Functions (IRF) for the Gambia: (**a**) is the response of GDP to an impulse of Tourism, (**b**) is the response of GDP to an impulse of HC, (**c**) is the response of GDP to an impulse of NC and (**d**) is a response of HC to an impulse of NC.

The model for Trinidad and Tobago, as can be seen in Figure 2, was estimated at the lag length of one, as the two lags proposed by the Akaike [26] information criterion again resulted in estimation errors. The impulse from a shock in Tourism provides a slightly positive response, which has a value of close to 0 after 4 periods. The initial response of economic growth from HC is statistically insignificant, with a slight negative pattern from one to four periods. The result of an impulse of NC on economic growth is initially negative with definitive negative trend that persists up to five periods. The impact of NC on HC is initially slightly positive, and it starts to fluctuate after two periods after which it soon reaches limes 0 and becomes statistically insignificant.

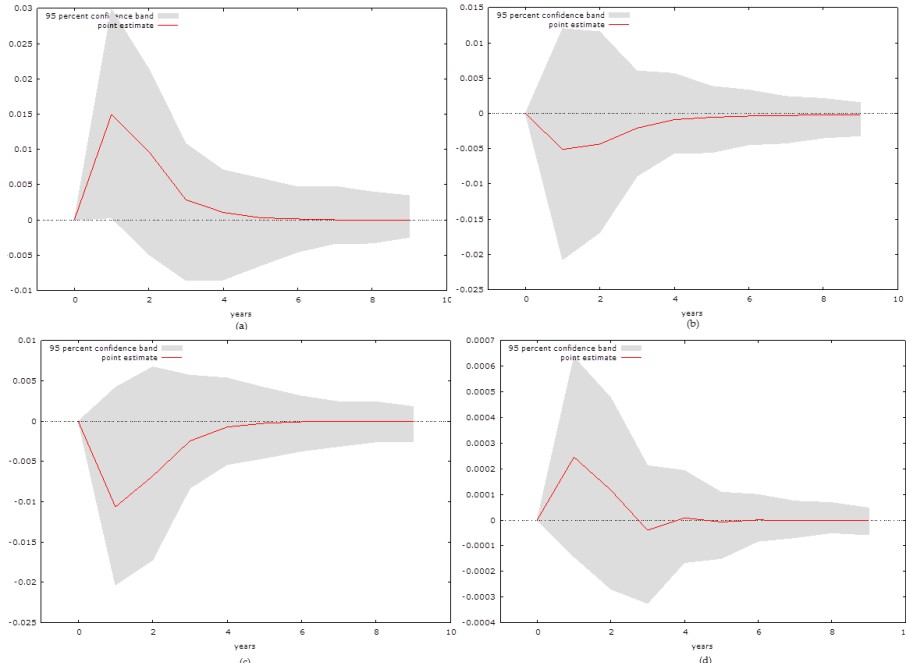

**Figure 2.** IRF for Trinidad and Tobago: (**a**) is the response of GDP to an impulse of Tourism, (**b**) is the response of GDP to an impulse of HC, (**c**) is the response of GDP to an impulse of NC and (**d**) is a response of HC to an impulse of NC.

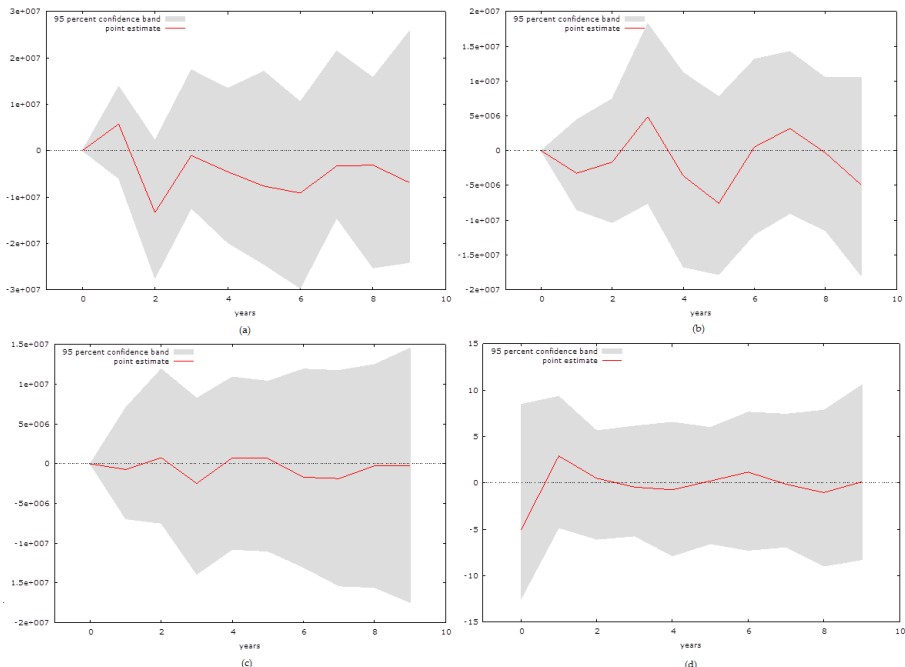

**Figure 3.** IRF for Lesotho: (**a**) is the response of GDP to an impulse of Tourism, (**b**) is the response of GDP to an impulse of HC, (**c**) is the response of GDP to an impulse of NC and (**d**) is a response of HC to an impulse of NC.

Lesotho, which is to a degree dependent upon natural resources, displays different tendencies, which can be seen in Figure 3. The model was specified at the lag length two. It clearly has a positive short-term impact of Tourism on GDP that starts to fluctuate after two periods and the results continue

to fluctuate, with negative trends in the long-run. This is logical, as tourism is not a significant determinant of the GDP of Lesotho. The initial response of economic growth to HC is negative, while the long-term response seems to be statistically insignificant. The result regarding natural capital conforms to the overall hypothesis of the natural resource curse, as the initial response is slightly positive, yet most of the results seem to be negative. The response of NC to HC seems to be statistically insignificant, as the initial result is negative, but it starts to fluctuate without a definitive pattern after two periods.

The VAR model for Belize, which can be seen in Figure 4, includes one lag of the variables, as there are estimation errors with the two lags proposed by the Akaike information criterion. The response of GDP to Tourism appears to be negative and this trend persists throughout the first eight periods after which it becomes neutral. This is especially interesting when taking into account that the economy of Belize is highly dependent upon tourism. The results of the VAR framework suggest that HC causes moderately negative impulses in the first eight periods. The results are even more conclusive in the case of NC where it seems that there is a slight negative impulse followed by stronger negative impulses from three to nine periods, after which the impulses become positive. The impact of NC on HC is statistically insignificant, with an initial impulse, that is limes 0 and slightly negative responses persist throughout the observed periods.

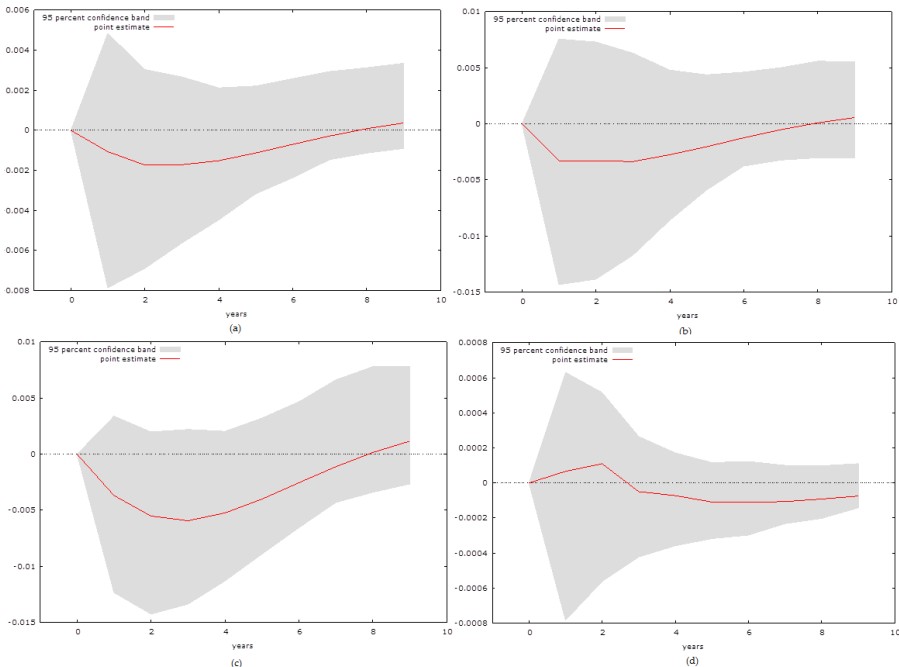

**Figure 4.** IRF for Belize: (**a**) is the response of GDP to an impulse of Tourism, (**b**) is the response of GDP to an impulse of HC, (**c**) is the response of GDP to an impulse of NC and (**d**) is a response of HC to an impulse of NC.

The results for Guyana, which can be seen in Figure 5, differ significantly to all of the other natural resource-dependent countries. This issue will be addressed in detail in the discussion. Regarding the empirical results, at this point it is sufficient to state that at the lag length of one, the effect of tourism on GDP is slightly positive and the positive trend persists throughout all of the observed periods. The impact of human capital on economic growth is negative and the negative trend persists. The impact of natural capital differs significantly to all of the other observed natural resource-dependent economies. The initial response is positive and all of the responses remain positive in the long-term. The results of the impact of HC on NC are initially negative, while they seem to become positive in the long-run.

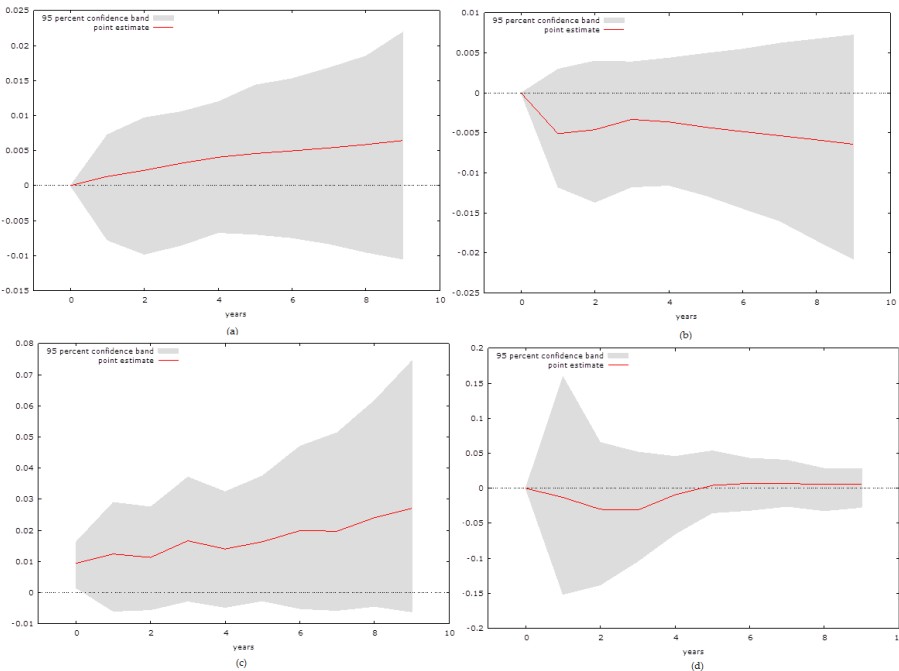

**Figure 5.** IRF for Guyana: (**a**) is the response of GDP to an impulse of Tourism, (**b**) is the response of GDP to an impulse of HC, (**c**) is the response of GDP to an impulse of NC and (**d**) is a response of HC to an impulse of NC.

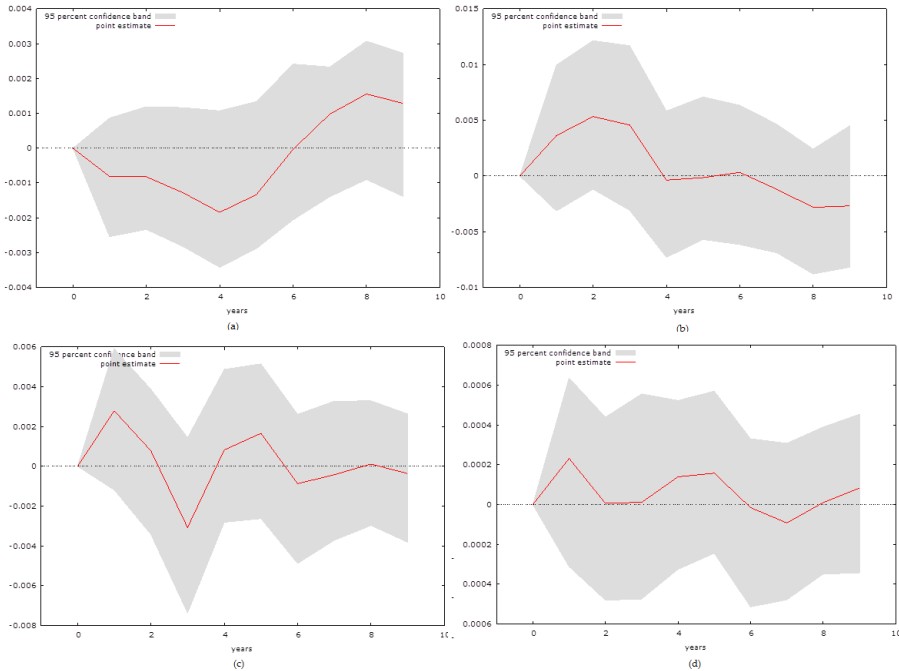

**Figure 6.** IRF for Jamaica: (**a**) is the response of GDP to an impulse of Tourism, (**b**) is the response of GDP to an impulse of HC, (**c**) is the response of GDP to an impulse of NC and (**d**) is a response of HC to an impulse of NC.

Jamaica is also estimated at the lag length of two, where all of the diagnostic tests confirm the stability of the observed models. The results can be observed in Figure 6. Interestingly, as was the case in the Belize IRF's, tourism seems to cause a significant negative response in GDP that persists for seven periods. This is interesting as both of these economies are dependent upon tourism revenue.

The response of economic growth to NC and HC fluctuates on both accounts where there are positive initial signs and fluctuations in the midterm. The initial response of HC to NC seems to be neutral with mostly positive responses from two to eight responses and negative in the long term.

Fiji appears to have some similarities to other tourism-dependent economies. The IRFs of the VAR model, shown in Figure 7, are estimated at the lag length one. Primarily, there is a negative impact in economic growth caused by tourism. This impact was initially negative and most of the remaining observed periods are negative until the responses become statistically insignificant. The response of economic growth to HC is initially negative and the responses mostly seem to persist throughout the observed 10 periods. The initial response of GDP to NC is initially positive, yet the results strongly fluctuate after four periods without a definitive trend. This differs significantly from the results of Jamaica and Belize.

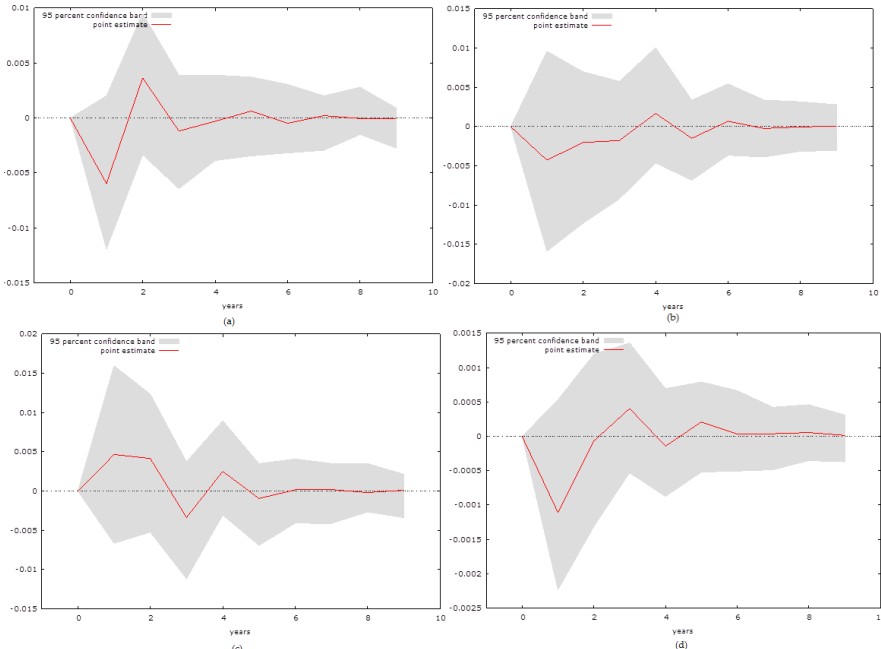

**Figure 7.** IRF for Fiji: (**a**) is the response of GDP to an impulse of Tourism, (**b**) is the response of GDP to an impulse of HC, (**c**) is the response of GDP to an impulse of NC and (**d**) is a response of HC to an impulse of NC.

The results for Liberia, displayed in Figure 8, show some similarities to the previous models. Liberia is not a country that is significantly dependent upon tourism, as it accounts for roughly two percent of its GDP, so the lack of a response from tourism towards economic growth is logical. There is only an initial weak positive response from HC towards economic growth, while in the middle and long term there is no significant relationship. NC causes several negative impulses in the first several periods, yet they appear to be insignificant in the long-term period. The relationship between NC and HC seems to be statistically insignificant.

The economy of Malawi has some different traits in comparison to other resource-dependent economies which can be observed in Figure 9. Primarily, Malawi is dependent upon agriculture as a source of economic growth, which is far less prone to external shocks. The initial response of economic growth to tourism is negative and such a trend persists throughout the first six periods. The impulse of HC on economic growth provides a slightly negative response, although the value becomes close to being statistical insignificant after six periods. The impact of NC on economic growth is positive in the short run, but reaches limes 0 after three periods. The initial response of HC to NC is positive and there is a definitive positive long-term trend, which seems to decrease in the intensity of its significance as the number of impulses increases.

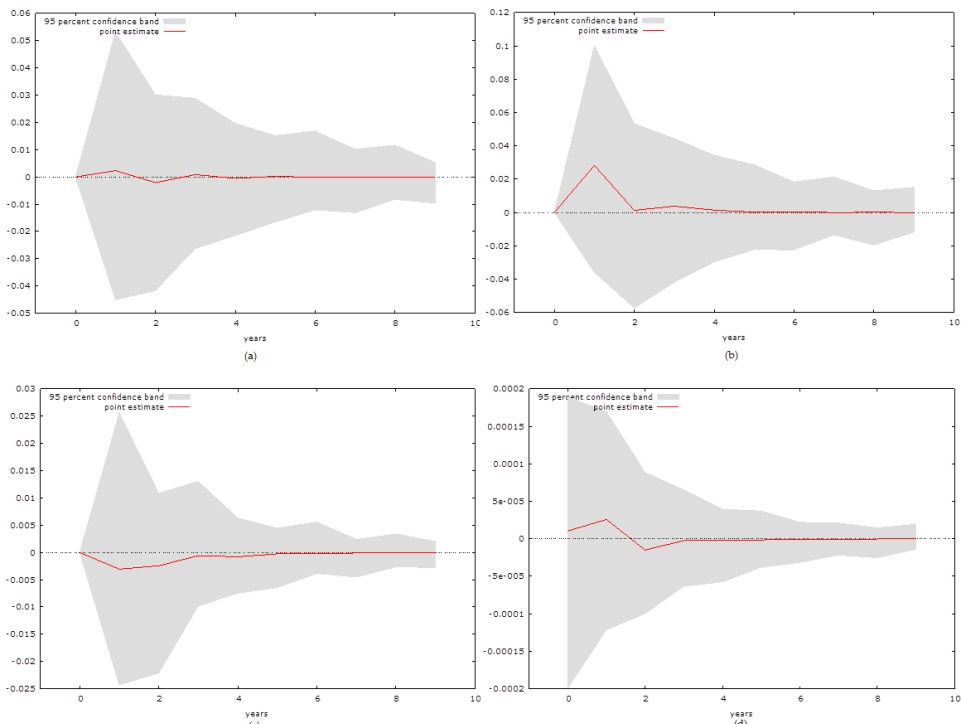

**Figure 8.** IRF for Liberia: (**a**) is the response of GDP to an impulse of Tourism, (**b**) is the response of GDP to an impulse of HC, (**c**) is the response of GDP to an impulse of NC and (**d**) is a response of HC to an impulse of NC.

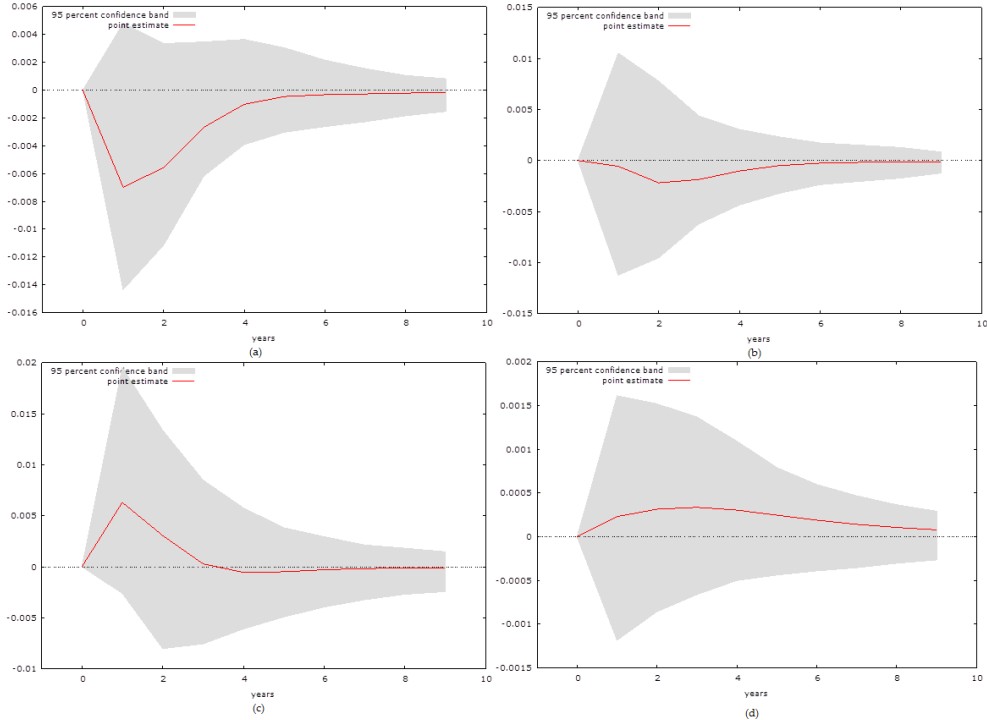

**Figure 9.** IRF for Malawi: (**a**) is the response of GDP to an impulse of Tourism, (**b**) is the response of GDP to an impulse of HC, (**c**) is the response of GDP to an impulse of NC and (**d**) is a response of HC to an impulse of NC.

The results for Barbados have some similarities with the other small economies dependent upon tourism, as can be seen in Figure 10. Primarily, this refers to the negative response of GDP to tourism, where there is a persistent negative trend. This seems to suggest that tourism revenue, up to a certain degree, may constrain economic growth. There are several obvious methods of transmission, such as the difficulty of diversifying the economy if the economy is actually so highly dependent upon tourism. The response of economic growth to HC is positive and the positive responses persist throughout the all ten observed periods. The results of the responses of both economic growth to NC and HC to NC seem to be statistically insignificant. Both of the results fluctuate without a definitive pattern.

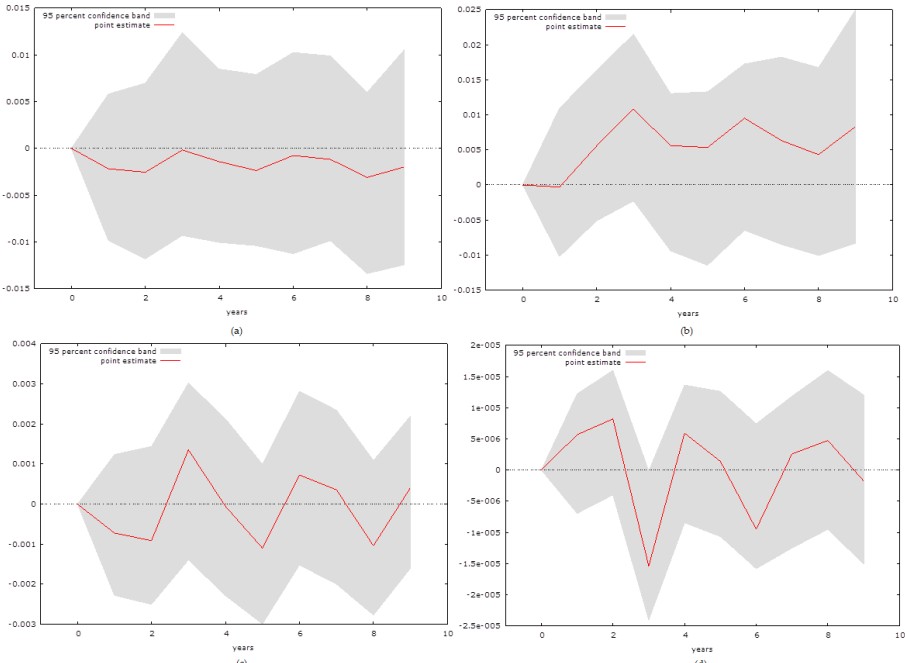

**Figure 10.** IRF for Barbados: (**a**) is the response of GDP to an impulse of Tourism, (**b**) is the response of GDP to an impulse of HC, (**c**) is the response of GDP to an impulse of NC and (**d**) is a response of HC to an impulse of NC.

## 4. Discussion

The discussion is divided into three sections. The first deals with the interpretation of the results gained from the VAR model regarding the natural resource-dependent economies. The second deals with the discussion regarding the tourism-dependent economies, while the third section discusses in detail theoretical aspects related to the concepts of the resource curse and human capital, respectively.

### 4.1. Discussion of Analysed Natural Resource-Dependent Economies

The first aspect of the discussion focuses on the countries that might be under influence of the natural resource curse. The first thing that should be noted in countries is that the negative influence of natural capital is more pronounced in countries that are (1) heavily dependent upon oil and (2) do not achieve diverse exports in neither partners they trade with nor the products they export. This is mostly visible in the case of Trinidad and Tobago, where there is a clear case of a strong negative impulse caused by NC. There are two reasons for this. Primarily, the economy of Trinidad and Tobago is roughly 65 percent dependent upon the export of various forms of oil [24], (Artana 2007 [34]). This would confirm the oil curse, as oil is a commodity whose value constantly fluctuates on the international market and the economy of the countries which produce it can do very little to prepare for such fluctuations. While larger producers have a say in OPEC, the previously stated is especially true for small economies that do not have significant political influence.

The second strong negative impact is for the IRF for Lesotho, although the effect is less pronounced than in Trinidad and Tobago. The production of Lesotho is perhaps more diversified than in the case of Trinidad and Tobago, but the problem with Lesotho is that it mainly produces commodities that are available from other exporters, such as wool and mohair, while perhaps the resource it is best known for are its diamond deposits and limited amounts of other minerals [35]. None of these commodities are essential and most of them can be substituted with products from other exporters and any negative spike in the global market decreases demand for most of the products Lesotho exports, making it highly prone to shocks in the international financial market.

There is the evidence of an initial negative response for Liberia, though there seems to be evidence of a strong response that exists in Trinidad and Tobago and Lesotho. Unlike Lesotho and Trinidad and Tobago, Liberia is dependent on exports of goods such as iron ore, rubber, wood and cocoa beans [36], (Beevers, 2012 [37]). Based on this information, we conclude that Liberia's exports are far more diversified than the exports of Trinidad and Tobago. It should be noted that most of the resources it exports are necessary for the functioning of modern industrial countries, unlike the products that Lesotho exports. It should also be noted that the prices of iron ore do not fluctuate negatively as the price of crude oil, as from 1986 to 2008 the price of iron ore has remained mostly stable with only two significant positive fluctuations in 2004 and 2008 [38]. As was previously emphasized, these economies do not have a strong position in determining the price and negotiating, which overall conforms to the conclusions of Braveboy-Wagner (2010 [39]).

The last two natural resource-dependent economies that are considered are Malawi and Guyana. The short-term positive impact and long-term neutral impact in Malawi is probably because Malawi, as an economy, is mostly dependent upon agriculture. Because the majority of its population is also employed in that sector and it is far less dependent upon exports, it means that it is far less prone to shocks in the international economy. In addition, its average TNRR value is roughly eight percent, meaning that overall its economy is not completely dependent upon natural resources. Guyana is a case where if we accept the "necessity of a diverse export" hypothesis we should expect a highly negative effect of NC on economic growth. Despite the fact that Guyana is dependent upon the export of one resource almost as Trinidad and Tobago is on petroleum, the short-term effect of NC is highly positive, with a persistent positive trend.

This is because roughly 58 percent of Guyana's export is dependent upon gold [40]. Gold has been, for a long time, a very stable resource whose value has fluctuated very little in the past 30 years. There have even been claims that gold is a safe-haven used in times of crisis, a claim investigated by Baur and McDermott (2010 [41]). During the times of crisis not only does the price of gold not decrease, but actually increases, as it is perceived as a commodity whose price is not prone to fluctuation, confirming that even 45 years after the US President Nixon abolished the gold window, gold is still perceived as a stable commodity. Perhaps an interesting case can be made that the natural resource curse is highly dependent upon which resource the economy is dependent upon, at least considering empirical evidence from five small economies observed in this paper. The key questions, which determine the properties of these countries, seem to be: (1) how stable are the prices of the resources these countries are dependent on; and (2) how hard is it to substitute these products.

The general conclusions regarding small natural resource-dependent economies are:

(1) The negative impact of NC on economic growth is more evident in countries that are dependent upon the export of commodities whose prices are significantly influenced by shocks in the international market;

(2) Generally, more diverse economies tend to have a smaller negative impact of NC on economic growth, as well as economies that export resources whose prices are less prone to shocks in the international market;

(3) There is enough empirical evidence of the oil curse in the case of Trinidad and Tobago;

(4) The negative effect is more pronounced in countries with a larger average TNRR, unless the commodity they are dependent on has a highly stable price and is difficult to substitute.

*4.2. Discussion of Analysed Tourism-Dependent Economies*

The difficulty with measuring the impact of natural capital in tourism-dependent economies is that it does very little to measure the actual strain that increased tourism activity may cause to a particular environment. The difficulty of the majority of existing variables that are used to measure natural capital is that in the majority of the analysed countries their natural capital is the factor that is attracting tourists. On average, these countries do seem to have a larger value of GVA than the natural resource-dependent countries, such as Jamaica, which has a certain amount of natural resources, mainly bauxite [42]. The most that can be done is to interpret these results taking into account the limitations of the variables.

The main measurable indicator of the impact of financial capital was tourism as a percentage of GDP and in the economies that are most dependent upon tourism and other external activities it would appear that, the majority of the impulses in the observed ten periods were negative. To some extent, the exception was the Gambia, where the first impulse of impulses of tourism seemed to cause neutral responses with only a slight negative initial impulse. This is perhaps because the Gambia is not completely dependent upon tourism, as it depends upon tourism for roughly 20 percent of its GDP [33]. Even more important, among the five tourism-dependent economies we analysed, the Gambia is by far the most dependent on foreign aid and has the lowest living standard, as well as being highly dependent on foreign aid [43]. Reports and other research suggests that tourism helps the Gambia combat its high level of poverty (Mitchell and Faal, 2007 [44]). As such, in the case of a country where the GDP per capita was $528 [24], the scale and continuation of foreign aid is not so affected by shocks in the international market.

The case is different when examining Belize, Barbados, and Jamaica, where most of the impulses of tourism are negative. Carneiro (2016 [45]) found that the economy of Belize is highly prone to external threats and faced significant decline in the aftermath of the recent crisis, as well as in the case of the recession in the United States in the early 1990s. More et al. (2015 [46]) have also emphasized the fact that the economy of Barbados is highly dependent upon energy imports, especially oil, making them additionally prone to shocks in the international market.

Narayan (2004 [47]) found that an increase in tourist revenue in Fiji of 10 percent causes a roughly 0.5% increase in real GDP. Fiji has perhaps an economy that is more diverse and equally under strain from internal shocks (coups and episodes of political violence), as well as external shocks, making it a highly specific example. Taking the results for Fiji with caution, as it is equally as dependent on tourism as Belize and Barbados, and even more so than Jamaica [33], we still find that there is empirical evidence of tourism causing negative impulses in GDP in the majority of tourism-dependent economies. This is highly interesting taking into account the fact that in the small economies we have observed that the most significant form of capital (in regards to economic growth) in the countries we have observed, produces negative impulses in the economic growth of these economies.

The impact of HC on economic growth produces mostly negative or neutral impulses in all of the ten observed economies, with the clear exception of Barbados. Perhaps more significantly, it is clear that in the least developed economies of those observed—the Gambia, Lesotho, and Malawi—we find no evidence of positive impulses caused by HC. This conforms to the general hypothesis of several researchers that human capital has a more significant effect in more developed economies (Neagu, 2012 [48]; Akpolat, 2014 [49]). In all of the observed economies we have failed to establish that, NC has a long-term significant impact on HC. In most cases, we have observed that the initial response is positive, but the constant fluctuation that usually occurs after two or three periods means that we fail to establish a statistically constant and significant long-term relationship. We further fail to find exact evidence that there is a negative link going from NC towards economic growth in tourism-developed counters. There are some strong differences in this area, as the impulses strongly fluctuate in the cases of the Gambia and Barbados, while they appear to be positive for Fiji and strongly negative for Belize. It should be noted that of the fluctuating patterns noted in the Gambia and Barbados, most of the

impulses are negative. Further research is necessary and we would advise most of it to be directed towards establishing variables that would more precisely measure HC and NC.

*4.3. Theoretical Discussion Regarding the Natural Resource Curse and Human Capital*

The long-term stagnation of growth or negative growth, as well as the fluctuations of growth, experienced by resource-dependent countries, can be pinned to the resource curse, and fluctuations of prices of commodities on the world market, respectively [1]. Small economies can escape the resource curse because small size of the market (in terms of land area and population) may lead to less diversification of raw materials and resources, which restricts domestic production (Castello and Ozawa, 1999 [50]). On the other hand, small island developing states (SIDS) tend to be particularly prone to exogenous shocks such as natural disasters, international political instability and fluctuations in prices of raw materials. Despite this, the idea of vulnerability should be considered in the context of the degree to which economies manifest resilience in tackling shocks (Camilleri and Falzon, 2013 [51]). This is also evident in our research, as the more diverse economies seem to have a lesser negative impulse from their most dependent form of capital.

"In the last 40 years, there is a negative robust correlation between the share of resource exports in GDP and economic growth. This correlation remains even when many other factors are controlled for. The robustness of the correlation between resource abundance and growth, even when we control for many other factors, gives us an indication that there may be a causal effect from resources to growth—but only an indication. And this is the main challenge of the empirical literature on the resource curse as it now stands" ([8], p. 244). This is a significant challenge in many research papers that have managed to prove correlation, but have failed to establish a causal link.

"It has been proven that natural resource abundant economies tend to grow slower than economies without substantial resources. ( . . . ) Nevertheless, many growth winners such as Botswana, Canada, Australia, and Norway are rich in resources. Moreover, of the 82 countries included in a World Bank study [52], five countries belong both to the top eight according to their natural capital wealth and to the top 15 according to per capita income" (Mehlum, et al., 2006 [53], p. 1).

In an extensive review of the empirical and theoretical literature, van der Ploeg [7] analyses and provides evidence of popular hypotheses explaining the channels through which resources can negatively affect the economic performance of nations or regions (and thereby causing the resource curse): "The best available empirical evidence suggests that countries with a large share of primary exports in GNP have bad growth records and high inequality, especially if quality of institutions, rule of law, and corruption are bad. This potential curse is particularly severe for point-source resources such as diamonds and precious metals." ([7], p. 406)

It is especially important to point out the negative influence of poor institutions, corruption, and anticipation of better times, which is directly connected with negative genuine savings.

Mehlum, et al. [53] emphasize the importance of institutions as channels for the development or evasion of resource curse. Poor institutions, marked by ineffective oversight, politically motivated rent allocation, and high levels of corruption, among other features, mean that a country (usually poor and resource-dependent) is captured by resource curse. The authors study the average yearly economic growth from 1965 to 1990 versus resource abundance in a total of 42 studied countries that have more than 10% of their GDP as resource exports. The indication of a resource curse only appears for countries with inferior institutions, while the indication of a resource curse vanishes for countries with better institutions. Institutions may be decisive for how natural resources affect economic growth even if resource abundance has no effect on institutions. Botswana, with 40% of GDP stemming from diamonds, has had the world's highest growth rate since 1965. Natural resources put the institutional arrangements to a test, so that the resource curse only appears in countries with inferior institutions.

The corruption level is also directly connected with the quality of institutions and the nature of the regime. Therefore, the resource rich countries with low levels of corruption of the public sector are the only ones that can escape the resource curse.

Atkinson and Hamilton ([10], pp. 1794–1795) have confirmed that their main indicator of resource abundance, the share of resource rents in GDP, is negatively correlated with the GDP per capita growth rate. In analysing the wider relationship between the resource curse, savings and growth, it was found that those resource abundant countries that have suffered from a curse appear to be those countries that have low or negative genuine savings. Pearce and Atkinson (1993, [54]) provided one of the earliest suggestions for a practical indicator of (weak) sustainability based on this proposition. This adjusted national savings measure accounts for the depletion of natural resources (and the environment). Hamilton (1994 [55]; 1996 [56]) developed welfare measures to account for living and non-living resources and varieties of pollutants in a similar extended national accounting framework—-the corresponding (net) savings measure, equivalent to that of Pearce and Atkinson, was termed "genuine" saving, and it measures the extent to which countries are, on balance, liquidating or creating national wealth.

Genuine saving, defined as above, if it is negative, is exactly the right indicator of resource curse, hence all the countries analysed in some of the aforementioned studies [7,10,11] show this connection, having negative genuine saving in the analysed periods, when an intensive exploitation of natural resources has either begun and continued or simply went on occurred: "Countries with a large percentage of mineral and energy rents of GNI typically have lower genuine saving rates" ([7], p. 396).

Consequently, not a single country analysed ([8], p. 246) that has managed to escape the path of resource curse has a negative genuine savings rate. The research was based on resource-adjusted savings rates as percentage of gross national income, average 1972–2000. Among the countries listed as escapers, 10 out of 11 have positive resource-adjusted savings rates (and it is questionable whether the one that has not—Oman—really has escaped the resource curse).

If it wants to have any chance of escaping the resource curse, a society faces a new allocation challenge: how much natural capital should be converted to economic production, and how much should be conserved for the provision of ecosystem services? Both uses of natural capital are essential and have no substitutes (Farley, 2012 [57]). Although it is obvious, beyond doubt, that GDP and economic growth is improved by liquidating natural capital (Naidoo, 2004 [58]). However, the real question is: should the natural capital be transformed into some other form of capital before leaving the country, in order to maintain a higher and long-term GDP growth? The proponents of the resource curse thesis mentioned before have no doubts about it. For a country that has natural capital, it is better to transform it into some other form of capital before exporting it. In resource extraction dependent countries, natural capital is usually exported without its transformation, and in most cases, resource curse is a consequence. In tourism-dependent countries, natural capital is used as a means to attract tourists (mostly foreign). In that way, services are exported and natural capital does not leave a country, hence it is of a different kind (landscapes, natural environments etc.).

Human capital—the learning, abilities, skills, and knowledge of an individual—can be used in the labor market as a form of currency (or capital) in exchange for wages or earnings. Human capital is often considered a key predictor of a person's employment and wages. Human capital theory (Voora and Venema, 2008 [59]) suggests that investments in human capital can be through formal schooling or on-the-job training, both of which raise workers' productivity and therefore increase their wages or earnings.

Due to a lack of estimates for this variable in less developed countries, it has not been possible to assess the importance of this determinant for their growth and development (Moreira, et al., 2014 [60]). Past and present literature has suggested that human capital is a key socio-economic development tool (Nam, et al., 2010 [61]; Quisumbing and McNiven, 2010 [62]). Countries are encouraged to expand and develop a robust educational system, yet many of these countries face barriers to development through human capital (Moore and Daday, 2010 [63]). Thus, we conclude that although there is a significant amount of literature dealing with human capital, there are clear examples of research voids and a lack of consensus on the measurement of these variables.

## 5. Conclusions

Based on our theoretic and empirical analysis of the available data on small tourism and natural resource-dependent economies, we have managed to draw several significant conclusions. Primarily, natural capital seems to produce negative impulses in the majority of resource-dependent economies, especially those that are dependent upon resources that are easily substituted or have prices that are highly prone to influence from external shock. It is our belief that of the five observed natural resource-dependent economies, the only exception is Guyana. We believe that this is because the economy of Guyana is largely dependent upon gold, which is very difficult to substitute and its price is mostly stable.

When observing similarities and differences between the resource-dependent and tourism-dependent countries, we find that natural capital does not significantly negatively influence economic growth in the tourism-dependent economies. We believe that this is partially due to the limitation of the variable used to measure natural capital, which does very little to measure the long-term damaging effects tourism can have on the environment. We find one significant similarity, the type of capital, which influences economic growth in both sets of countries primarily, produces negative impulses when viewing the results of the IRF functions. The only exceptions that we found were countries that either had a highly diverse economy or had resources whose price was relatively stable in the international market and thus less prone to shock.

The main policy recommendation based on the results of our empirical analysis is that countries should attempt to diversify their economy by focusing on more than one key determinant of economic growth. Focusing on only one key segment of the economy a strategy that involves a significant amount of risk, especially the volatility of the products in the international market and possible shocks that these small economies cannot predict nor can they in any way compensate for the occurrence of such shocks. Thus, as our empirical research suggests, developing a more diverse economy significantly reduces the potential risks of foreign shock, which is highly important for the economies that we have observed.

**Author Contributions:** Petar Kurečić developed the theoretical framework, concept and research arguments of the paper; Filip Kokotović developed the empirical framework of the paper. Both authors wrote the manuscript and approved the final version of the manuscript.

**Conflicts of Interest:** The authors declare no conflict of interest.

## Appendix A

As specified in the methodological section, we provide full results of the ADF unit root tests. The tests are conducted until the level where they reject the null hypothesis of non-stationarity at the 5% significance level. The specification of the test is with a constant and with no trend present.

**Table A1.** Unit root tests.

| Country | GDP | Tourism | HC | NC | GDPpc |
|---|---|---|---|---|---|
| The Gambia | −1.073 (0.782) | −3.548** (0.0068) | −0.134 (0.9943) | −1.895 (0.6566) | −2.803 (0.0507) |
| In first difference | −4.647** (0.0001) | / | −0.308 (0.9214) | −10.79** (0.0000) | −4.436** (0.00025) |
| In second difference | / | / | −7.628** (0.0000) | / | / |
| Trinidad and Tobago | −2.086 (0.2502) | −2.119 (0.237) | 1.153 (0.9979) | −4.678** (0.0075) | −2.138 (0.8004) |
| In first difference | −3.693** (0.0035) | −3.04* (0.031) | −1.356 (0.5795) | / | −3.097* (0.0361) |
| In second difference | / | / | −4.309** (0.0043) | / | / |
| Lesotho | 1.316 (0.9967) | −1.962 (0.2992) | −1.436 (0.5661) | −3.683** (0.0048) | 1.214 (0.9969) |
| In first difference | −3.509* (0.03829) | −4.903** (0.0011) | −0.767 (0.8043) | / | −4.115** (0.0059) |
| In second difference | / | / | −4.032** (0.0072) | / | / |

**Table A1.** *Cont.*

| Country | GDP | Tourism | HC | NC | GDPpc |
|---|---|---|---|---|---|
| Belize | −6.128** (0.0000) | −3.325** (0.0092) | 0.977 (0.9891) | −4.239** (0.0023) | −6.267** (0.0000) |
| In first difference | / | / | −1.232 (0.6364) | / | / |
| In second difference | / | / | −4.2549** (0.005) | / | / |
| Guyana | −2.694 (0.076) | −3.467** (0.0089) | −2.765 (0.2251) | −4.785** (0.0062) | 0.744 (0.9871) |
| In first difference | −4.507** (0.0011) | / | −1.205 (0.6481) | / | −4.012** (0.0073) |
| In second difference | / | / | −7.354** (0.0000) | / | / |
| Jamaica | −1.451 (0.8685) | −3.383* (0.0093) | −2.392 (0.1566) | −2.223 (0.2049) | −2.261 (0.1848) |
| In first difference | −5.176** (0.0000) | / | −8.781** (0.0000) | −5.283** (0.0000) | −4.986** (0.0001) |
| Fiji | −1.985 (0.2936) | −2.049 (0.2655) | −1.442 (0.563) | −1.623 (0.4604) | −1.262 (0.6496) |
| In first difference | −7.948** (0.0000) | −6.087** (0.0000) | −1.712 (0.4182) | −5.83** (0.0000) | −7.896** (0.0000) |
| In second difference | / | / | −6.325** (0.0000) | / | / |
| Liberia | −0.9836 (0.7144) | −1.15 (0.649) | −1.541 (0.5421) | −2.234 (0.194) | −0.7063 (0.708) |
| In first difference | −9.479** (0.0000) | −4.205** (0.0013) | −4.253** (0.00007) | −10.744** (0.0000) | −4.759** (0.0002) |
| Malawi | 1.039 (0.9971) | −2.583 (0.093) | −1.847 (0.3578) | −3.067* (0.0456) | 0.056 (0.9624) |
| In first difference | −3.741* (0.0125) | 4.9047** (0.0000) | −1.032 (0.5521) | / | −4.137 (0.0053) |
| In second difference | / | / | −4.237** (0.0046) | / | / |
| Barbados | −2.767 (0.0817) | −2.695 (0.0747) | −1.218 (0.6687) | 1.696 (0.997) | −2.83 (0.0728) |
| In first difference | −3.146* (0.041) | −4.025** (0.0071) | −5.523** (0.0000) | −5.9101** (0.0000) | −3.142* (0.041) |

Source: Authors' Calculations and GRETLE output Note: values in the parenthesis represent the *p* value. * and ** indicate statistical significance at the respected 0.05 and 0.01 levels of significance.

The diagnostic tests regarding the VAR models are presented in Table A2, where the results confirm that there is no presence of the ARCH effect and no serial correlation of the residuals in the relevant equations. We present the individual ARCH and autocorrelation results for the equations that are relevant rather than the average value of all of the equations, but there is no presence of autocorrelation or the ARCH effect in any of the observed models.

**Table A2.** Diagnostic tests.

| Country | Autocorrelation GDP equation | Autocorrelation HC equation | ARCH GDP equation | ARCH HC equation |
|---|---|---|---|---|
| The Gambia | 0.34 (0.56) | 0.155 (0.866) | 0.008 (0.929) | 0.162 (0.779) |
| Trinidad and Tobago | 0.454 (0.5) | 1.502 (0.22) | 0.319 (0.572) | 1.766 (0.183) |
| Lesotho | 0.8505 (0.356) | 0.124 (0.899) | 0.549 (0.4587) | 0.266 (0.623) |
| Belize | 0.925 (0.336) | 0.943 (0.331) | 0.742 (0.3889) | 0.701 (0.403) |
| Guyana | 1.641 (0.201) | 0.021 (0.804) | 0.181 (0.671) | 0.966 (0.325) |
| Jamaica | 1.005 (0.316) | 0.988 (0.321) | 2.307 (0.1287) | 0.8591 (0.354) |
| Fiji | 0.2456 (0621) | 0.227 (0.633) | 0.144 (0.704) | 0.093 (0.7599) |
| Liberia | 0.2516 (0.616) | 1.099 (0.294) | 0.148 (0.7003) | 1.0054 (0.3159) |
| Malawi | 0.421 (0.431) | 0.187 (0.821) | 0.625 (0.428) | 0.055 (0.814) |
| Barbados | 0.583 (0.445) | 0.196 (0.658) | 0.079 (0.7788) | 0.324 (0.569) |

Source: Authors' Calculations and GRETLE output. Note: values in the parenthesis represent the *p* value.

The Cholesky ordering is presented in Table A3. We have experimented with changing the ordering of NC and tourisms in the model where GDP is the main variable, but found the changes to be negligible.

**Table A3.** Cholesky ordering for the IRFs for natural resource dependence.

| Dependent variable | Cholesky ordering |
| --- | --- |
| GDP | GDP, GDPpc, Tourism, NC, HC. |
| HC | HC, GDPpc, GDP, Tourism, NC. |

The stability of the models is confirmed in Figure A1, where it can be seen that all of the inverse roots, represented by the blue dots, are within the bounds of the unit circle.

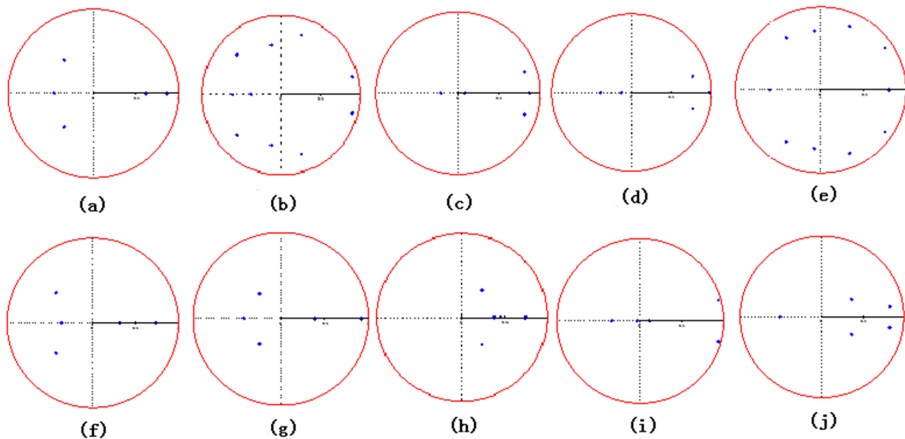

**Figure A1.** VAR Inverse Roots; as follows: (**a**) The Gambia, (**b**) Trinidad and Tobago, (**c**) Lesotho, (**d**) Belize, (**e**) Guyana, (**f**) Jamaica, (**g**) Fiji, (**h**) Liberia, (**i**) Malawi and (**j**) Barbados.

## Appendix B

The Gambia is a country that has an increased reliance upon tourism, as it depends upon tourism for approximately 20% of its GDP [26]. In both sets of countries we have included countries that are neither completely dependent upon tourism, nor on natural resources, but a high portion of their respected economies still depend on this sector. This was done in order to observe whether there were any differences between the countries that were completely dependent upon a particular segment of their economy in comparison to those that were less dependent on either natural resources or tourism. In the case of the countries dependent upon tourism, the Gambia was that country as it is a relatively poor country with a low GDP per capita and it is highly reliant on foreign aid. The remaining economies are clearly highly dependent upon tourism as a source of revenue, which can clearly be seen from Figure A2, based on WTCT data [26].

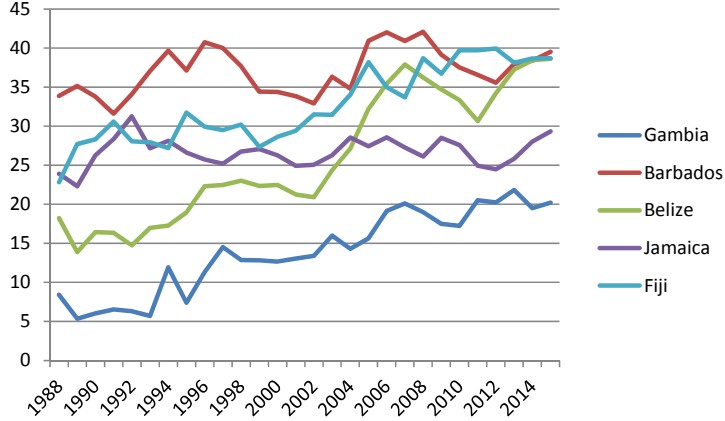

**Figure A2.** Total tourism revenue as a percentage of GDP.

The countries from the group of natural resource-dependent countries all highly rely on the export of various minerals or on resources as sources of revenue or they are dependent upon agriculture, such as Malawi. In Figure A3 we provide the plotted figures of the total natural resource rent, based upon World Bank data [17].

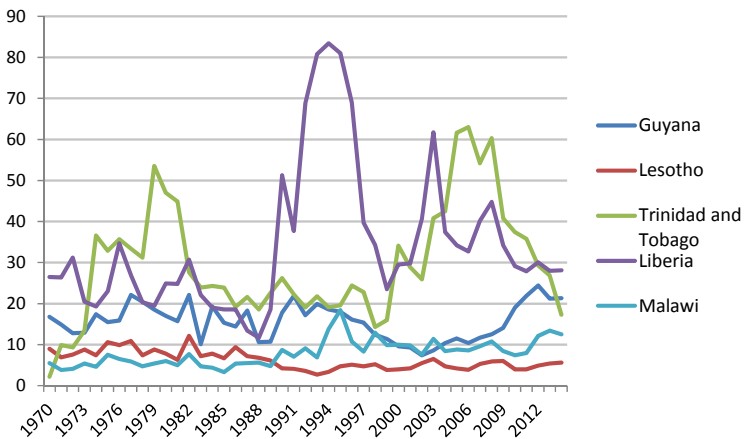

**Figure A3.** Total natural resource rent as a percentage of GDP.

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
