# Peer review of "Examining the "Natural Resource Curse" and the Impact of Various Forms of Capital in Small Tourism and Natural Resource-Dependent Economies"

_economies, doi:10.3390/economies5010006_

Round 1

Reviewer 1 Report

In my opinion the paper presents major shortcomings and shold be deeply revised to become suitable for publication. Below I list few major points to provide the authors with a starting point for revision.

A clear theoretical background of the proposed empirical expercise is missing. The growth models discussed in section 2 do not provide any theorethical hypotheses on how the economy's dependence on the exploitation of natural resources should reduce their future growth. A broad stream of literature (not cited in the paper) should be discussed and may give a more suitable thoretic ground to the paper:a s a starting point I suggest (Atkinsons et Hamilton 2003; Hamilton and Atkinsons, 2006; Torvik, 2009; Van Der Ploeg 2011).

It is completely unclear to me how WB estimates on natural resource rents can  represent the dependence of small economies from tourism in those countries where the exploitation on non-renewable natural resurces is negligible. Why the authors didn't include a variable to represent the dependence from tourism by itself (f.i. share of tourism on GDP)? Moreover, while the dynamic impact of the exploitation of non-renewable resources on growth can be modeled (see the cited literature), completely unclear are the possible causal relationship linking the current dependence from tourism and the future growth path.

Page 4 row 145 and page 11 row 393. The authors assume that TNRR variable expresses  the stock of natural capital while the WB estimate refers to annual exploitation of non renewable natural resource. (to be used as a correction term to calculate the genuine saving indicator of sustainability). In no meaningful sense the TNRR variable (either expressed as a % of GDP or in absolute value) can be used as a measure of natural resource total endowment (stock) of a given economy, being simply a measure of dependence of the current economic performance on the exploitation of nonrenewable natural resources.

On the econometric side, in a dynamic context I am concerned about possible problems of endogeneity: look at the relationship between GDP and GFCF.

References

Atkinsons, G. and Hamilton, K. (2003) ‘Savings, growth and the resource curse hypothesis’, World Development, Vol. 31, No. 11, pp.1793–1807.

Hamilton, K. and Atkinson, G. (2006). Wealth, welfare and sustainability: advances in measuring sustainable development. Edward Elgar Publishing.

Torvik, R. (2009) ‘Why do some resource-abundant countries succeed while others do not?’, Oxford Review of Economic Policy, Vol. 25, No. 2, pp.241–256.
Van der Ploeg, F. (2011) ‘Natural resources: curse or blessing?’, Journal of Economic Literature, Vol. 49, No. 2, pp.366–420.

Author Response

We thank the reviewers for their very useful comments. We have taken all of the suggestions into account.

The variable, which measures natural capital is now Gross Value Added (for further information, see section 4.3. of the paper) and tourism (as percentage of GDP) is now included as the measurement of the financial capital that is observed in the paper instead of Gross Fixed Capital Formation.

We find most of our research results are valid despite the change in proxy variables, but where there are differences we have addressed and rewritten parts of the discussion. The use of tourism helps decrease potential endogeneity issues, although as stated in the paper, no variable is completely exogenous.

All of the research papers which have been proposed by the reviewers are now included in the paper and we welcome these helpful inclusions to the paper.

The introduction and theoretic discussion of the paper are now completely rewritten to comply with the suggestions of the reviewers. All the literature suggestions were studied and the most important points were used in order to address the reviewers' objections.

.  

Reviewer 2 Report

Interesting paper with potential to influence the literature on this topic. however, I have several observations that the authors must take in consideration in order to improve their manuscript for publication:

- The discussion about economic growth model and theory in the initial pages and formulas is irrelevant and just a waste space in the paper... this is a well known discussion that does not need to be put here

-  Show in table 2 the time period from where this average is coming from

- Table 3 is totally redundant 

- Review your figures... add axis names, change scale so they are readable,     

- Many countries have 'escaped' the RC, so it is worth to discuss when this islikely to happen or not... see the references below 

Finally, the authors are missing key literature and discussion about the resource curse or the effect of mining expansion. For instance, see the following:

https://www.aeaweb.org/articles?id=10.1257/jel.49.2.366 --a MUST read/cite to understand the causes fo the RC

http://onlinelibrary.wiley.com/doi/10.1111/1467-8489.12118/full --a must to contextualize theRC in 'small' economies

http://www.tandfonline.com/doi/abs/10.1080/00049182.2015.1020596 --to better understand the link resources to income inequality

Author Response

We thank the reviewers for their very useful comments. We have taken all of the suggestions into account.

The variable, which measures natural capital is now Gross Value Added (for further information, see section 4.3. of the paper) and tourism (as percentage of GDP) is now included as the measurement of the financial capital that is observed in the paper instead of Gross Fixed Capital Formation.

We find most of our research results are valid despite the change in proxy variables, but where there are differences we have addressed and rewritten parts of the discussion. The use of tourism helps decrease potential endogeneity issues, although as stated in the paper, no variable is completely exogenous.

All of the research papers which have been proposed by the reviewers are now included in the paper and we welcome these helpful inclusions to the paper.

The introduction and theoretic discussion of the paper are now completely rewritten to comply with the suggestions of the reviewers. All the literature suggestions were studied and the most important points were used in order to address the reviewers' objections.

Round 2

Reviewer 2 Report

No more comments